# Identification of NCAPG as an Essential Gene for Neuroblastoma Employing CRISPR-Cas9 Screening Database and Experimental Verification

**DOI:** 10.3390/ijms241914946

**Published:** 2023-10-06

**Authors:** Yubin Jia, Jiaxing Yang, Yankun Chen, Yun Liu, Yan Jin, Chaoyu Wang, Baocheng Gong, Qiang Zhao

**Affiliations:** 1Department of Pediatric Oncology, Tianjin Medical University Cancer Institute and Hospital, Tianjin 300060, China; jiayubin@tmu.edu.cn (Y.J.); tjykd796@outlook.com (J.Y.); 15165189816@163.com (Y.C.); elvahb@126.com (Y.L.); jinyan19900713@163.com (Y.J.); wangcy@tmu.edu.cn (C.W.); 2National Clinical Research Center for Cancer, Tianjin Medical University Cancer Institute and Hospital, Tianjin 300060, China; 3Key Laboratory of Cancer Prevention and Therapy, Tianjin 300060, China; 4Tianjin’s Clinical Research Center for Cancer, Tianjin 300060, China

**Keywords:** neuroblastoma, Depmap, CRISPR-Cas9, *NCAPG*, risk signature, proliferation

## Abstract

Neuroblastoma is the most common extracranial solid tumor in children. Patients with neuroblastoma have a poor prognosis. The development of therapy targets and the ability to predict prognosis will be enhanced through further exploration of the genetically related genes of neuroblastoma. The present investigation utilized CRISPR-Cas9 genome-wide screening based on the DepMap database to determine essential genes for neuroblastoma cells’ continued survival. WGCNA analysis was used to determine the progression-related genes, and a prognostic signature was constructed. The signature gene, *NCAPG*, was downregulated in neuroblastoma cells to explore its impact on various cellular processes. This research used DepMap and WGCNA to pinpoint 45 progression-related essential genes for neuroblastoma. A risk signature comprising *NCAPG* and *MAD2L1* was established. The suppression of *NCAPG* prevented neuroblastoma cells from proliferating, migrating, and invading. The results of flow cytometric analysis demonstrated that *NCAPG* inhibition caused cell cycle arrest during the G2 and S phases and the activation of apoptosis. Additionally, *NCAPG* downregulation activated the *p53*-mediated apoptotic pathway, inducing cell apoptosis. The present work showed that *NCAPG* knockdown reduced neuroblastoma cell progression and may serve as a basis for further investigation into diagnostic indicators and therapy targets for neuroblastoma.

## 1. Introduction

Neuroblastoma is the prevailing non-cerebral solid neoplasm observed in pediatric patients. The initiation of neuroblastoma occurs during the maturation of the sympathetic nervous system [1]. Neuroblastoma constitutes roughly 15% of the total fatalities resulting from cancer in the pediatric population. The median presentation age is 23 months [2]. The patients with low- and intermediate-risk neuroblastoma have a favorable prognosis, with 5-year survival rate of 97.9± 0.5% and 95.8 ± 0.8%, respectively. However, the 5-year survival rate of high-risk neuroblastoma patients was only 62.5 ± 1.3% despite having received multimodal treatment [3]. It has been reported that neuroblastoma is clinically severe depending on tumor characteristics, including histology, stage, classification, and tumor cytogenetics [4]. The initiation and development of neuroblastoma are underpinned by crucial genetic factors, including but not limited to *MYCN* amplifications, *MDM2* amplifications or overexpression [5], *ALK* mutations or amplifications, and segmental chromosomal abnormalities [6]. The development of therapy targets and the ability to predict prognosis will be enhanced through further exploration of the genetically related genes of neuroblastoma.

High-throughput screening projects such as DepMap, which utilizes RNAi silencing and CRISPR-Cas9, have provided a practical approach for identifying potential dependency genes essential for tumor survival [7,8]. The Cancer Dependency Map (https://depmap.org/portal/, accessed on 10 October 2022) is a publicly available online platform that utilizes extensive multiomics screening initiatives [9], including the Cancer Cell Line Encyclopedia (CCLE) [10], the PRISM Repurposing dataset employing pooled-cell line chemical-perturbation viability for small molecule screening [11], and the Achilles Project, which relies on genome-scale CRISPR-Cas9 knockout screens [12]. The DepMap offers gene dependency information for over 700 human tumor cell lines with diverse tissue origins and information on gene expression, gene copy number, and gene mutation for over 1,000 tumor cells. Through the DepMap database, we can obtain the gene dependency data of neuroblastoma cell lines. Based on the DepMap database analysis, we identified *NCAPG* as a possible treatment target for neuroblastoma. The *NY-MEL-3* gene encodes the *NCAPG* (Non-SMC condensin I complex subunit G), located on the 4p15.32 chromosome of the human genome. During meiotic and mitotic cell division, *NCAPG* compacts and stabilizes chromosomes [13,14]. By regulating cell aging, cell cycle, and mismatch repair, *NCAPG* may modulate tumor occurrence and development [15]. In addition, *NCAPG* has been observed to facilitate the growth, migration, and invasion of diverse tumor cells, including hepatocellular carcinoma [16,17], prostate adenocarcinoma [18], breast invasive carcinoma [19], gastric carcinoma [20] via TGF-β and PI3K/AKT signaling. But, further research is needed to understand how *NCAPG* is expressed in neuroblastoma and its role in biology.

This study used the CRISPR-Cas9 database and neuroblastoma GSE49710 dataset (Zhang, W. et al., 2015) to screen neuroblastoma progression-related genes via WGCNA and LASSO analyses [21]. Then, a prognostic signature based on these genes was constructed. Lastly, to assess the *NCAPG*’s oncogenic potential, the effects of knocking it down on cell proliferation and apoptosis were further evaluated.

## 2. Results

### 2.1. Identification of Candidate Genes That Were Crucial for Neuroblastoma Survival

From DepMap data, genome-wide CRISPR-based loss-of-function assays were utilized to identify essential gene candidates implicated in neuroblastoma. The DepMap database thoroughly analyzed the two factors of gene copy number and sgRNA loss while utilizing CRISPR-Cas9 technology to evaluate the dependence of cells on genes and defined a new parameter, CERES, as a parameter to estimate the degree of gene requirement (Meyers, R.M. et al. 2017). The CERES principle states that the fewer cells that carry the gene’s sgRNA survive, the fewer copies of the gene there in the cell, the more the cells depend on the gene [12]. Using a CERES score of <−1 in 75% of neuroblastoma cell lines, 605 genes were recognized as required for neuroblastoma (Appendix A). In addition, the GSE49710 dataset was subjected to the Weighted correlation network analysis (WGCNA) analysis to ascertain the genes linked to the progression of neuroblastoma (Appendix A). WGCNA is a systems biology method used to describe the gene association patterns between different samples. It can be used to identify gene sets with highly synergistic changes and to identify candidate biomarker genes or therapeutic targets based on the internal connectivity of gene sets and the association between gene sets and phenotypes (Langfelder, P. & Horvath, S. 2008) [22]. The brown modules (correlation parameter = 0.42, *p* < 0.001) and turquoise modules (correlation parameter = −0.46, *p* < 0.001) were found to be correlated with neuroblastoma progression (Figure 1A,B). The brown and turquoise modules revealed 236 hub genes, and 45 genes intersected in hub genes and essential genes (Figure 1C, Appendix A).

We performed enrichment analysis to explore the potential biological functions of these genes in neuroblastoma. KEGG enrichment analysis of these 45 genes indicated that they are most heavily enriched in the following areas: the cell cycle, DNA replication, nucleotide excision repair, oocyte meiosis, the mismatch repair pathway, *P53* signaling pathways, pyrimidine metabolism, glutathione metabolism, and base excision repair pathways. Furthermore, GO enrichment results revealed that these genes were primarily associated with catalytic activity, DNA binding activity, protein serine kinase activity, helicase activity, chromosome, centromeric region, DNA replication, chromosome segregation, and nuclear division (Figure 1D). The findings indicated that the essential genes may play a potential role in cell cycle, DNA replication, and *P53* signaling in neuroblastoma.

Subsequently, theSTRING (https://cn.string-db.org/, accessed on 18 October 2022) database and Cytoscape software (Version 3.10.0) were employed to construct a network of protein–protein interactions (PPI) involving the 45 crucial genes (Figure 1E). MCODE can identify densely connected regions in vast protein–protein interaction networks that may represent molecular complexes. The technique relies on vertex weighting via local neighborhood density and outward traversal from a locally dense seed protein to extract the dense regions following supplied parameters (Bader, G.D. et al. 2003) [23]. In addition, ten genes, including *NCAPG*, *RAD51*, *MAD2L1*, *AURKB*, *BUB1B*, *NDC80*, *CDC6*, *DTL*, *KIF11*, and *RRM2*, were predicted as being hub genes in neuroblastoma using the MCODE—algorithm (Figure 1F).

### 2.2. The Development and Verification of a Gene-Based Prognostic Signature

To further explore the hub genes involved in the progression of neuroblastoma, the GSE49710 dataset was partitioned into two datasets, namely the training dataset and the internal validation dataset, using a random split with a ratio of 7:3. We performed the LASSO-COX analysis (Tibshirani R. 1997) among these 10 genes to screen genes for risk signature construction in the training dataset, and two genes and corresponding regression coefficients were identified in the analysis (Figure 2A,B) [24]. Subsequently, the risk score was calculated as the sum of the product of each signature gene expression and its corresponding regression coefficient (The risk score = the *NCAPG* expression × 0.562 + the *MAD2L1* expression × 0.416). For the training and internal validation dataset, high-risk and low-risk groups were identified using the median risk score as a cut-off value. Based on the Kaplan–Meier survival analyses conducted on the internal training and validation datasets, it was observed that the low-risk group exhibited a more favorable prognosis in patients diagnosed with neuroblastoma (Figure 2C and Appendix A). Figure 2D and Appendix A displayed neuroblastoma patients’ survival status and risk scores. The prognosis of neuroblastoma patients in internal training datasets and internal validation datasets could be accurately predicted by the risk score, according to the analysis of the ROC curve (Figure 2E and Appendix A).

The validation of the prognostic signature was extended by utilizing the E-MTAB-8248 dataset sourced from the ArrayExpress database. The preceding prognostic signature calculated a risk score for each individual. The media risk score divided the patients into high-risk and low-risk cohorts. Patients with neuroblastoma within the low-risk group had better survival rates than those within the high-risk group, according to an examination of the Kaplan–Meier curve (Appendix A). The distribution of risk scores and survival status for patients with neuroblastoma is shown in Appendix A. The risk signature employed in our study demonstrated high accuracy in predicting neuroblastoma outcomes (Appendix A).

The current risk classification system, stage and *MYCN* status have been widely recognized as neuroblastoma prognostic factors. Therefore, the prognostic value of this risk signature in children with unfavorable clinical characteristics may be more meaningful. We investigated the prognostic value of the risk signature in samples with age > 18 months, *MYCN* amplification, progression, COG high-risk group, and INSS stage 4 in the training dataset. The high-risk group in the risk signature still presented with a significantly poor prognosis in patients with age >18 months, progression, COG high-risk group, and INSS stage 4 (Appendix A). Regrettably, the prognostic value of the risk signature was not significant in the *MYCN*-amplified subgroup, and the low sample size may be an important reason (Appendix A). We also verified this result in the internal validation dataset and external validation dataset. The risk signature showed significant prognostic value in patients with unfavorable characteristics, including age >18 months and INSS stage 4 (Appendix A).

### 2.3. NCAPG Expression Correlated with Clinical Outcomes and Clinical Characteristics

Median *NCAPG* expression levels were used to divide patient samples from the GSE49710 dataset into high and low-expression groups. Patients with neuroblastoma who expressed *NCAPG* highly had a worse chance of surviving (Figure 3A). The univariable and multivariable analysis examined the impact of patient characteristics such as gender, age, *MYCN* status, and INSS stage on the correlation between *NCAPG* expression and prognosis (Figure 3B,C). The findings of the study indicated a significant association between the expression of *NCAPG* and the outcomes of patients with neuroblastoma.

The study conducted a correlation analysis between *NCAPG* expression and various clinical characteristics of neuroblastoma patients. The findings of the study demonstrated a statistically significant association between the expression of *NCAPG* and several clinical variables, such as age, *MYCN* status, clinical risk, INSS stage, and progression (Figure 3D). Neuroblastoma with amplified *MYCN* expressed higher *NCAPG* than neuroblastoma without amplified *MYCN* (Figure 3E). In addition, patients with high-risk versus low-risk neuroblastoma expressed considerably more *NCAPG* (Figure 3F). The rise in *NCAPG* expression and the patients’ rising INSS stage were found to be positively correlated. The relative expression of *NCAPG* was lower in individuals diagnosed with stage 4S than those diagnosed with stages 3 and 4 (Figure 3G). Neuroblastoma progression patients expressed significantly higher levels of *NCAPG* (Figure 3H).

### 2.4. Neuroblastoma Tissues Expressed High Levels of NCAPG

Immunohistochemistry was used to measure NCAPG protein expression levels in 40 neuroblastoma tissue samples (Figure 4A). NCAPG protein was highly expressed in 47.5% (19/40) of neuroblastoma tissues (Figure 4D). Clinicopathological parameters include gender, age, INSS stage, COG risk, bone marrow metastasis, distant metastasis, histological type and *MYCN* amplification (Appendix A). A notable association was detected between the upregulation of NCAPG protein and several clinical factors, including the INSS stage (*p* = 0.004), COG risk (*p* = 0.004), and distant metastasis (*p* = 0.027) (Table 1). The Kaplan–Meier curves showed that neuroblastoma patients who had highly expressed NCAPG had poorer EFS and OS (*p* = 0.0138 and 0.0213, respectively) (Figure 4B,C).

The correlation between NCAPG protein high expression and OS time in neuroblastoma patients was analyzed using univariate and multivariate methods to establish its independent prognostic significance. The results of the univariate analysis demonstrated a statistically significant correlation between the increased expression of NCAPG and unfavorable overall survival in patients diagnosed with neuroblastoma (*p* = 0.033). Moreover, INSS stage (OS, *p* = 0.009) and distant metastasis (OS, *p* = 0.003) showed a significant influence on poor prognosis. According to multivariate analysis, distant metastasis in neuroblastoma was significantly associated with shorter OS (*p* = 0.045) time (Table 2). Therefore, NCAPG protein overexpression could be considered a prognostic biomarker for neuroblastoma.

### 2.5. NCAPG Regulated Neuroblastoma Cells Proliferation, Migration, and Invasion

The assessment of NCAPG expression was conducted in SH-SY5Y, SK-N-BE (2), SK-N-SH, and SK-N-AS cells using qRT-PCR and Western blot analyses. NCAPG was expressed at elevated levels in SH-SY5Y and SK-N-BE (2) cells (Figure 4E,F). Silencing the *NCAPG* gene in SH-SY5Y and SK-N-BE (2) confirmed its significance in neuroblastoma. We used Western blotting and qRT-PCR to identify *NCAPG* knockdown in SH-SY5Y and SK-N-BE (2) cells transfected with control shRNA or shRNAs targeting *NCAPG*. The *NCAPG* mRNA and protein levels significantly decreased following sh2-*NCAPG* and sh3-*NCAPG* transfected in SH-SY5Y and SK-N-BE (2) cells (Figure 4G–J).

As a next step, we looked at *NCAPG*’s biological role in neuroblastoma cells. The results of the CCK-8 assays demonstrated a noteworthy reduction in cellular proliferation at 24, 48, and 72 h in SH-SY5Y and SK-N-BE (2) cells after the silencing of *NCAPG*, in contrast to cells that were transfected with shRNA control (Figure 5A,B). *NCAPG* downregulation inhibited the colony formation abilities of neuroblastoma cells in colony formation assays (Figure 5C–E). The function of *NCAPG* in migration and invasion was examined using transwell experiments in which SH-SY5Y and SK-N-BE (2) cells had reduced *NCAPG* expression levels. It was shown that SH-SY5Y and SK-N-BE (2) cells transfected with sh-*NCAPG* had a lower number of migrating cells than control cells. Moreover, neuroblastoma cells transfected with sh-*NCAPG* showed markedly reduced invasion ability (Figure 5F–I). The findings of these investigations provided evidence in favor of the conjecture that *NCAPG* functions as an oncogenic factor in human neuroblastoma cell lines, as its suppression impeded the cells’ ability to proliferate, migrate, and invade.

### 2.6. NCAPG Knockdown Caused Cell Cycle Arrest and Apoptosis in Neuroblastoma Cells

GSEA was conducted to explore the cellular roles of *NCAPG* in neuroblastoma. Based on GSEA results, the high expression of the *NCAPG* group was enriched in G2M CHECKPOINT. In contrast, the low expression of the *NCAPG* group was enriched in apoptosis (Figure 6A and Figure 7A). In *NCAPG* knockdown cells, the proportion of cells in the S and G2 stages of the cell cycle increased, accompanied by a decrease in the fraction of cells in the G1 phase (Figure 6B–E). The function of *NCAPG* in neuroblastoma cell apoptosis was examined using flow cytometry. As expected, both cell lines showed increased apoptosis rates in the sh-*NCAPG* group. In contrast, the knockdown of *NCAPG* promoted early apoptosis in SH-SY5Y cells, while both early and late apoptotic cells were significantly increased in SK-N-BE (2) cells (Figure 7C–F).

After that, we explored the molecular mechanisms that cause apoptosis in cells when *NCAPG* is knocked down. The results of the Gene Set Enrichment Analysis (GSEA) revealed that the group exhibiting low expression of *NCAPG* demonstrated significant enrichment in *P53* pathways (Figure 7B). The expression of p53, Bcl-2, Bax, Caspase9, and Caspase3 in neuroblastoma cells after *NCAPG* knockdown was detected using Western blotting. The results indicated an upregulation of p53, Bax, and Caspase9 and a downregulation of Bcl-2 and Caspase3 (Figure 7G,H). The study’s results suggested that the inhibition of *NCAPG* via shRNA interference prompted the activation of the apoptotic pathway mediated by *p53* in neuroblastoma cells.

## 3. Discussion

Neuroblastoma is a malignant neoplasm that arises from the embryonic neural crest cells of the autonomic nervous system, predominantly affecting pediatric patients during their developmental stages [2,25]. *MYCN* amplification, *TP53* deletions, *ALK* mutations and amplifications, *TERT* rearrangements, *ATRX* deletions, and segmental chromosomal aberrations are among the key genetic factors that contribute to the development of neuroblastoma [26,27,28]. Neuroblastoma is characterized by heterogeneous clinical presentations. When the diagnosis is made, more than half of all neuroblastoma patients are older, have an unresectable or metastatic illness, and are considered high-risk patients. Over half of these patients do not survive despite intensive, multimodal therapy [1,29,30].

The identification and understanding of gene functions and their correlation with the advancement of cancer may offer valuable insights into the development of cancer biomarkers and therapeutic interventions. The discovery of genes that could be potential candidates for involvement in cancer cell survival and proliferation is crucial [31]. Recently, the utilization of CRISPR-CAS9 screening has gained prominence as a potent instrument for precision medicine. It is particularly well-suited for identifying crucial genes associated with cancer cell survival and targeting cancer therapy. This approach facilitates the precise identification of genes critical for tumor cell survival and proliferation on a genomic scale [7,8]. Conducting a large-scale loss-of-function screening through RNAi or CRISPR/Cas9 techniques offers a distinctive perspective on the mechanisms by which human cancer cell lines sustain their fundamental immortality traits. Additionally, it enables the determination of whether a particular gene is indispensable for the cell’s proliferation and survival. The DepMap is the most extensive lentiviral-based genome-wide pooled CRISPR-Cas9 screen in the Avana 19Q2 collection, with studies of 17,634 genes covering 1714 cancer cell lines [32]. DepMap has created a computational technique (CERES) to evaluate gene dependency on the basis of essentiality screenings using CRISPR-Cas9 and consider copy-number-specific effects. The DepMap has enabled us to identify new potential biomarkers for neuroblastoma development.

The dependence score was computed via CERES using the DepMap website, and 605 genes were identified as indispensable for the proliferation and survival of neuroblastoma cells. Among the 605 genes, 45 were identified in neuroblastoma tissues associated with neuroblastoma progression through WGCNA analysis. According to the enrichment analysis, these 45 essential genes played major roles in neuroblastoma’s cell cycle and DNA replication. After that, we built a PPI network and singled out ten hub genes. We also developed and verified a risk signature incorporating the *NCAPG* and *MAD2L1* genes.

NCAPG protein is associated with chromosomal mitosis condensation and plays an imperative role in regulating cancer formation and growth [33]. According to current research, *NCAPG* exhibits a strong correlation with tumor size, histological grade, TNM stage, and OS, making it a promising biomarker for various cancer prognoses. *NCAPG* has been observed to impact the biological behavior of malignancies both in vivo and in vitro. Specifically, it has been discovered to facilitate tumor cell proliferation, migration, and invasion via pathways like TGF-β and PI3K/Akt [34]. However, *NCAPG*’s role in neuroblastoma is unclear.

The analysis of the GSE49710 dataset revealed a significant correlation between elevated levels of *NCAPG* expression and an unfavorable prognosis among individuals diagnosed with neuroblastoma. Moreover, it was observed that *NCAPG* exhibited significant upregulation in individuals with neuroblastoma who had *MYCN* amplification, high-risk, high-stage disease, and tumor advancement. NCAPG protein expression was assessed using immunohistochemistry on tumor specimens obtained from neuroblastoma patients for the present study. The findings indicated a significant association between NCAPG expression and patient prognosis. It was consistent with the results of the dataset. Furthermore, we confirmed the mRNA and protein expression of NCAPG in four distinct cell lines of neuroblastoma. High levels of NCAPG were expressed in SH-SY5Y and SK-N-BE (2) cell lines. Our study investigated the function of *NCAPG* in neuroblastoma by knocking down *NCAPG* expression in SH-SY5Y and SK-N-BE (2) cells with shRNA plasmid vectors. Neuroblastoma proliferation, invasion, and migration were decreased by inhibiting *NCAPG* expression. Additionally, *NCAPG* knockdown caused apoptosis and stopped the cell cycle in the G2 and S phases, according to flow cytometric analyses. In conclusion, it appeared that *NCAPG* functions as an oncogene during neuroblastoma development.

The nuclear transcription factor *P53* has been identified as a regulator of cell proliferation, which operates by triggering cell cycle arrest and apoptosis when exposed to a range of cellular stressors, including DNA damage, hypoxia, and oncogene activation. Several genes associated with apoptosis regulated transcriptionally by *p53* have been identified. The activation of oncogenes leads to the promotion of apoptosis by p53 in a sequential manner that involves transactivation of Bax, the release of cytochrome C from mitochondria, and activation of caspase-9, Caspase-3, -6, and -7 are then activated. Multiple death checkpoints, such as blocking p53 activity, Bcl-2 family members regulating mitochondrial function, and caspase inhibitors, can prevent p53-mediated apoptosis [35,36,37]. The current work examined how *NCAPG* downregulation affected the activity of the *p53* gene and the genes that it regulates, including Bax, Bcl-2, Caspase9, and Caspase3, in neuroblastoma cells. The transfection of cells with shRNA interference vectors targeting the *NCAPG* gene resulted in augmented expression of p53, Bax, and Caspase9 and reduced expression of Bcl-2 and Caspase3, as evidenced via Western blotting. Our findings elucidated a fundamental mechanism through which blocking *NCAPG* expression triggers *P53*-mediated neuroblastoma apoptosis.

Our findings could enhance the understanding of neuroblastoma’s molecular mechanisms and improve our ability to diagnose and treat the disease. Although *NCAPG* showed significant prognostic value in our study, its role in improving the established risk stratification system still needs to be verified in large-cohort clinical research. In addition, normal cells also require *NCAPG* to maintain their normal biological functions, which makes targeted therapy against *NCAPG* challenging, and it is crucial to find an appropriate therapeutic window. Our subsequent work would continue to concentrate on the possibility of *NCAPG* in neuroblastoma clinical management and further mechanistic investigations to explore potential therapeutic targets.

## 4. Materials and Methods

### 4.1. The Identification of Essential Genes for Neuroblastoma Cell Survival Based on the Large-Scale CRISPR-Cas9 Screening Database

There was an accessible website called The Cancer Dependency Map (https://depmap.org/portal/, accessed on 10 October 2022) based on large-scale multi-omics screening projects, which include the Cancer Cell Line Encyclopedia, the PRISM Repurposing dataset, and the Achilles Project. The Achilles Project employed the CRISPR-Cas9 methodology to systematically disrupt individual genes and ascertain potential genes that were pivotal for the sustenance of tumors. The CERES score was generated to estimate the viability effect of knocking out a gene for tumor cells in the database [12]. A gene’s score might be negative (indicating that its knockout would reduce the corresponding cell line survival) or positive (meaning its deletion might increase the cell line survival). The CERES score for neuroblastoma was downloaded from the Cancer Dependency Map database. We defined genes as essential in the study if they appeared in more than 75% of neuroblastoma cell lines with a CERES score <−1.

### 4.2. Acquisition of Data

The GSE49710 dataset’s mRNA expression profile and clinical data (Zhang W et al. 2015) were obtained from the Gene Expression Omnibus (GEO) repository [21]. The E-MTAB-8248 dataset (Roderwieser A et al. 2019) was downloaded from the ArrayExpress database [38]. The neuroblastoma cell line expression data were downloaded from the Cancer Cell Line Encyclopedia (CCLE) database (https://sites.broadinstitute.org/ccle/, accessed on 15 October 2022).

### 4.3. Weight Gene Coexpression Network Analysis

The current study used the “WGCNA” R package (Version, 1.72-1) to perform a weight gene coexpression network analysis (WCGNA) on the GSE49710 dataset to find hub genes that may be connected with progression. Initially, the degree of adjacency was computed for each pair of genes, followed by assessing the optimal soft threshold power utilizing the standard scale-free network. Utilizing the topological overlap matrix (TOM) mitigated the influence of extraneous associations and noise. The formation of modules was achieved by implementing a dynamic tree-cutting methodology using TOM-based dissimilarity. When cutting height = 210, minModuleSize = 10, a dendrogram with colored assignments was utilized to generate the clustering dendrogram of genes. A scale-free network was ensured by using a soft threshold power of 5. The calculation of gene significance (GS) and module membership (MM) was performed on modules, and genes within the module exhibiting GS values greater than 0.2 and MM values greater than 0.8 were designated as hub genes.

### 4.4. Enrichment Analysis

The potential biological functions of hub genes were explored through the utilization of the “clusterProfiler” R package (Version, 4.8.0), which conducted Kyoto Encyclopaedia of Genes and Genomes (KEGG) and Gene Ontology (GO) enrichment analysis [39,40,41]. The adjusted *p* value < 0.05 was considered statistically significant.

### 4.5. Construction of the PPI Network and Hub Gene Identification

The STRING database generated the protein–protein interactions (PPI) network, which was then displayed using the Cytoscape software (Version 3.10.0; http://www.cytoscape.org/, accessed on 18 October 2022). The Molecular Complex Detection (MCODE) algorithm implemented in the Cytoscape software was utilized to identify ten hub genes within the protein–protein interaction (PPI) network.

### 4.6. Development and Verification of the Prognostic Risk Signature

The GSE49710 dataset was partitioned into two groups, namely the training group and the internal validation group, using a random split with a ratio of 7:3. To create a neuroblastoma risk signature in the training group, we used the “glmnet” R packages (Version, 4.1-4) and “survival” R packages (Version, 3.3-1) to execute the least absolute shrinkage and selection operator (LASSO) analysis and COX regression analysis with the ten hub genes discovered in the PPI network. A risk score was calculated as the sum of the product of each signature gene expression and its corresponding regression coefficient [24,42,43]. The prognostic significance of the risk signature was assessed by performing a survival analysis of overall survival (OS) probability using the “survival” and “survminer” R packages (Version, 0.4.8). To evaluate the specificity and sensitivity of the risk signature, the “timeROC” R package (Version, 0.3) was used to create the receiver operating characteristic (ROC) curve.

### 4.7. Clinical Samples

Forty samples of neuroblastoma tumor tissue during the period of May 2014 through June 2019 were collected from Tianjin Medical University Cancer Institute and Hospital. Corresponding clinical and prognostic data of each sample were gathered. The current study was conducted in accordance with the principles outlined in the Declaration of Helsinki by the World Medical Association. Additionally, it received approval from the Research Ethics Committee of Tianjin Medical University Cancer Institute and Hospital (E20210027).

### 4.8. Immunohistochemistry (IHC)

In short, the histopathological sections, with a thickness of 4μm, were subjected to an overnight baking process at 60 °C, followed by deparaffinization and dehydration. Subsequently, we carried out antigen repair and inhibited endogenous peroxidase activity. These slices were then incubated in a damp container at 4 °C for 24 h with primary anti-NCAPG (1:100; 24563-1-AP, Proteintech, Wuhan, China) antibody, followed by 1 h at ambient temperature with the matching secondary antibodies. Finally, 5 min of diaminobenzidine (DAB) addition followed until a brown reaction product was seen. Leica light microscopes were used to acquire digital images. IHC sections were evaluated using ImageJ software (Version, 1.53s) to determine the dyed area and IOD values. Then, average optical density (AOD) = IOD/Area was calculated. This average optical density (AOD) was used to represent NCAPG staining intensity (indicating the relative level of NCAPG expression). The density of signals in tissue regions from five fields chosen randomly was quantified in a blinded fashion for subsequent statistical analysis. High expression was defined as AOD ≥ 0.3, and low expression was defined as AOD < 0.3.

### 4.9. Cell Lines and Cultures

We obtained human neuroblastoma cell lines from the ACTT (SK-N-BE (2), IMR-32, SH-SY5Y, and SK-N-AS). Ten percent fetal bovine serum (FBS, BI) and one percent penicillin/streptomycin (Gibco, Billings, MT, USA) were added to MEM/F12 media to keep SK-N-BE (2) and SH-SY5Y cells alive and well. SK-N-AS cells were cultured in DMEM supplemented with 10% FBS and 1% penicillin/streptomycin (Gibco), while IMR-32 cells were grown in MEM (with NEAA, Non-Essential Amino Acids). The cells were nurtured in a 37 °C, 5% CO_2_ incubator with humidity.

### 4.10. Lentivirus Transduction

The *NCAPG* knockdown plasmid (sh*NCAPG*) and control (sh*NC*) were ordered from TSINGKE (Beijing, China). The target sequence of *NCAPG* was 5′-CGGGCAGTGTTATCATGTATT-3′ for LV-sh*NCAPG*1, 5′-GCTATGCAGAAGCATCTTCTT-3′ for LV-sh*NCAPG*2, 5′-GGATCCCAAGGATGATGAATG-3′ for LV-sh*NCAPG*3. Puromycin selection was employed to establish stable cell lines. Western blotting and quantitative real-time PCR (qRT-PCR) were used to assess the knockdown’s efficacy.

### 4.11. Western Blotting

Protein samples were generated by lysing cells with RIPA lysis buffer (Solarbio, Beijing, China). The BCA protein detection kit was utilized to quantify the protein concentrations in the supernatants (Thermo, Waltham, MA, USA). After being separated with 8% SDS, the protein samples were transferred to PVDF membranes. Before overnight incubation at 4 °C with the primary antibody, the membranes were subjected to a 1 h incubation with 5% BSA. The membranes were then exposed to secondary antibodies for 60 min at room temperature, followed by three TBST rinses. The GelView 6000Plus system, produced by Biolight Biotechnology in Guangzhou, China, created the band images. The Western blotting procedure employed several primary antibodies, including NCAPG (24563-1-AP), procured from Proteintech (Wuhan, China). Additionally, p53 (2524), Bax (41162), Bcl-2 (15071), Caspase-9 (9508), and Caspase-3 (9662) were obtained from Cell Signalling Technology (Danvers, MA, USA).

### 4.12. Quantitative Real-Time PCR

The cells were subjected to RNA extraction using the SteadyPure Quick RNA Extraction Kit (Accurate Biology, Changsha, China) to obtain the total RNA. Reverse cDNA transcription was performed using a PrimeScriptTM RT kit (Takara, Asaz, Japan). Using SYBR Green qPCR Mix (Takara), we determined the precise amount of *NCAPG* expression. The *GAPDH* forwards and reverse primers were 5′-CTCACCGGATGCACCAATGTT-3′ and 5′-CGCGTTGCTCACAATGTTCAT-3′, respectively. The *NCAPG* forwards and reverse primers were 5′-CGCGTTGCTCACAATGTTCAT-3′ and 5′-ACAACTGGAATGCTCTGGATGTAACTC-3′, respectively. The 2^−ΔΔCt^ method was applied to ascertain the relative expression levels of *NCAPG*.

### 4.13. Cell Counting Kit 8 Assay

The rate of cell proliferation was assessed using CCK-8. The cells were seeded onto 96-well plates and kept in a culture environment at a constant 37 °C and 5% CO_2_. CCK-8 solution (Solarbio, China) was added to each well at 0, 24, 48, and 72 h and subsequently incubated at 37 °C for a duration of 2 h. Three measurements of the sample’s absorbance at 450 nm were taken for each piece using the microplate reader (Biotek Instruments Inc., Vinusky, VT, USA).

### 4.14. Colony Formation Assay

Cell proliferation was quantified through a plate cloning experiment. A six-well plate was seeded with 1000 cells at 37 °C and 5% CO_2_. Following a 14-day incubation period, the colonies were immobilized using a 4% paraformaldehyde solution and subjected to staining with a 0.1% crystal violet solution. Photographs were then taken of the colonies for further analysis. The colonies were enumerated using ImageJ software.

### 4.15. Transwell Assay

The cells were placed into the upper chamber of the 24-well transwell plates, which had a pore size of eight μm and contained a serum-free medium (Corning, New York, NY, USA). To conduct an invasion test, the upper compartments were seeded with Matrigel (BD Biosciences, Franklin Lakes, NJ, USA). After this, a total volume of 600 μL of the whole medium was transferred to the lower compartment. Non-migratory cells were eliminated from the sample after 24 h. The cells in the lower chamber were first fixed with a solution containing 4% paraformaldehyde, then stained with a solution containing 0.1% crystal violet, and finally analyzed using imaging software.

### 4.16. Flow Cytometry

The cells were cultured on 6-well plates for a duration of 72 h, with a cell density of 2 × 10^5^ cells per well. To detect apoptosis in cells, we used the Annexin V-FITC Apoptosis Detection Kit per the directions provided by the manufacturer (Yeasen, Shanghai, China). The BD FACSCanto II (BD, USA) was used to gather the data, and Flow Jo software (Version, v10.6.2) was used to analyze it. The flow cytometry gating strategy were described in detail in Appendix A.

### 4.17. Statistical Analysis

All statistical evaluations and analyses were performed using R software (4.3.1) and SPSS software (Version 26.0). This research examined the association between the manifestation of NCAPG protein and clinicopathological factors by applying the chi-square test and Fisher’s exact test. Log-rank tests and Kaplan–Meier survival plots were implemented to measure differences in survival status. To assess group differences, the ANOVA analysis was used. The results were obtained using three independent tests and presented as the Mean ± standard deviation (SD). *p* < 0.05 was the statistical threshold for significance.

## 5. Conclusions

We identify a gene associated with neuroblastoma survival as *NCAPG* using the DepMap database. The *NCAPG* gene plays an oncogenic role in neuroblastoma cells and comprises cell cycle and apoptosis. As a result of this study, we can gain new insight into the pathogenesis of neuroblastoma as well as potential therapeutic measures to prevent and treat this disease.

## Figures and Tables

**Figure 1 ijms-24-14946-f001:**
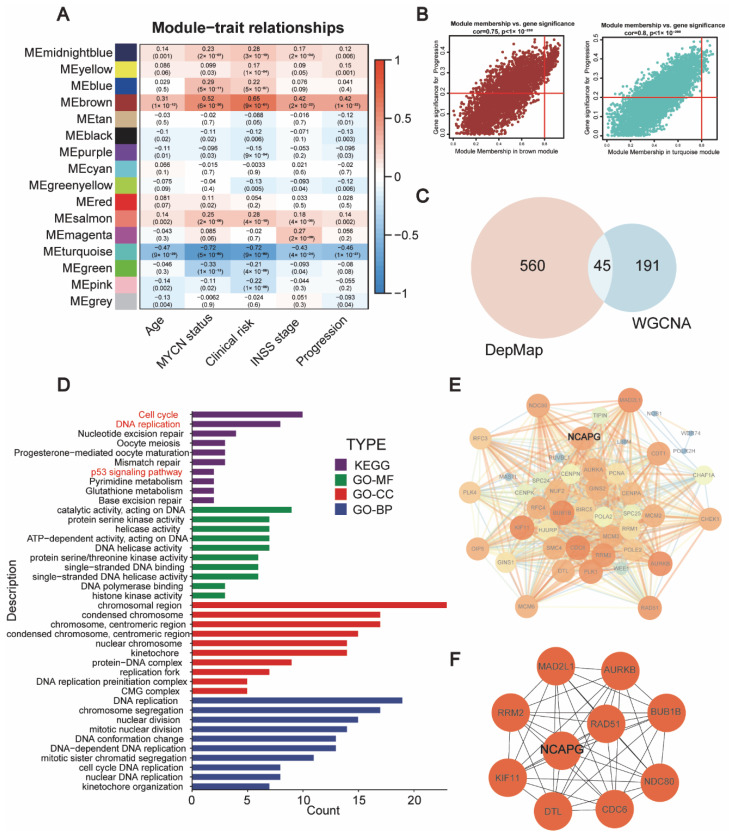
Identification of hub genes linked to neuroblastoma development using the Depmap and GEO datasets. (**A**) Heatmap of association of gene modules with clinical features. (**B**) Brown and turquoise module gene scatter plots. (**C**) Venn graph of overlaps in genes based on Depmap and WGCNA. (**D**) KEGG and GO analyses of 45 overlapped genes; these genes are most heavily enriched in the cell cycle, DNA replication and p53 signaling pathway. (**E**) PPI of 45 overlapped genes. (**F**) MCODE network clustering investigation of 45 overlapped genes.

**Figure 2 ijms-24-14946-f002:**
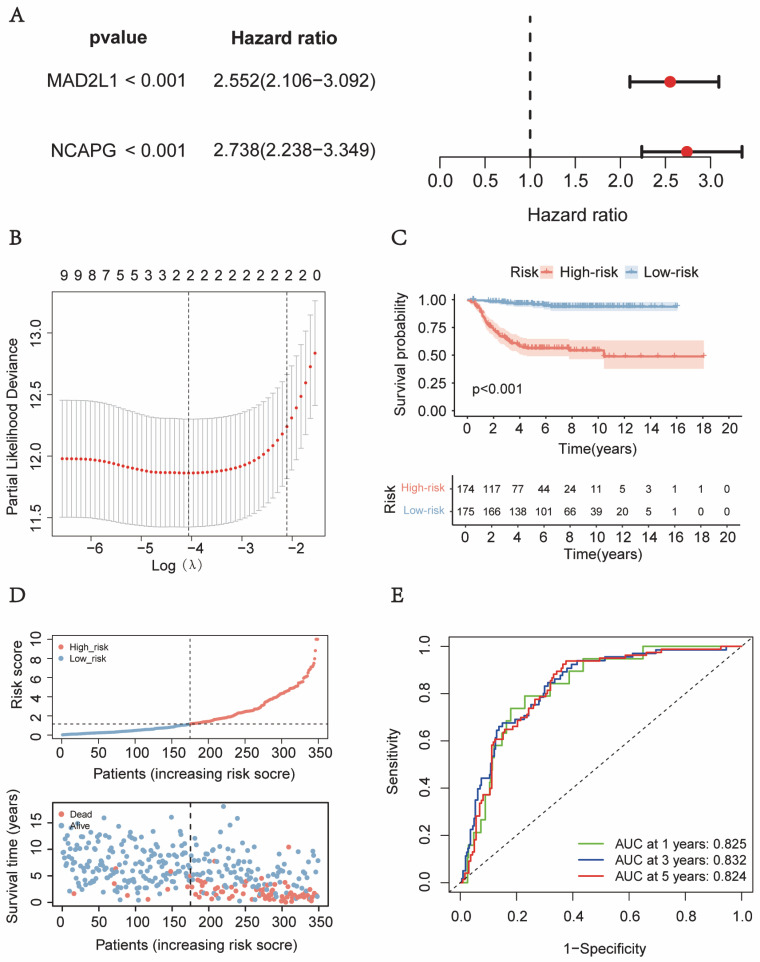
Creation of a neuroblastoma prognostic signature. (**A**) Univariate Cox analysis of survival for *MAD2L1* and *NCAPG*, the black dashed line represents the Hazard Ratio (HR) = 1, with HR greater than 1 categorizing the factor as a risk factor, and HR less than 1 categorizing it as a protective factor. (**B**) λ selection by 10-fold cross-validation, the red dotted line represents the partial-likelihood deviance estimated value and confidence interval of the corresponding log λ value in the LASSO analysis. (**C**) Plot showing the Kaplan–Meier distribution for the training cohort. (**D**) Patients in the train cohort’s risk scores and survival status distributions were displayed. (**E**) ROC curves for train cohort survival prediction, the black dashed line represents a scenario where the true positive rate is equal to the false positive rate. The larger the area under the ROC curve (AUC), which approaches 1, the stronger the diagnostic capability.

**Figure 3 ijms-24-14946-f003:**
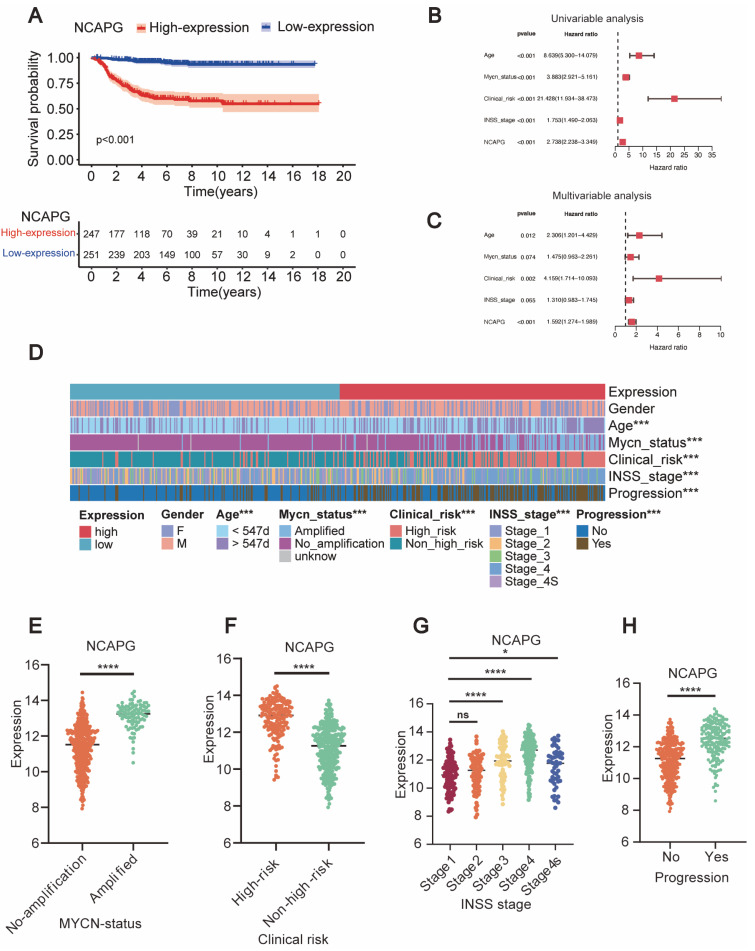
*NCAPG* was linked to a poor prognosis in neuroblastoma. (**A**) Neuroblastoma patients’ Kaplan–Meier survival curves for low- and high-expressed *NCAPG* in the GSE49710 dataset. (**B**,**C**) Univariable and multivariable analysis of *NCAPG* expression and patient survival in the GSE49710 dataset, the black dashed line represents the Hazard Ratio (HR) = 1, with HR greater than 1 categorizing the factor as a risk factor, and HR less than 1 categorizing it as a protective factor. (**D**) Relationship between neuroblastoma *NCAPG* expression and clinicopathological features in the GSE49710 dataset. (**E**–**H**) *NCAPG* expression differed depending on *MYCN* status, clinical risk, cancer grade, and progression in the GSE49710 dataset. ns: no significance, * *p* < 0.05, *** *p* < 0.001, **** *p* < 0.0001.

**Figure 4 ijms-24-14946-f004:**
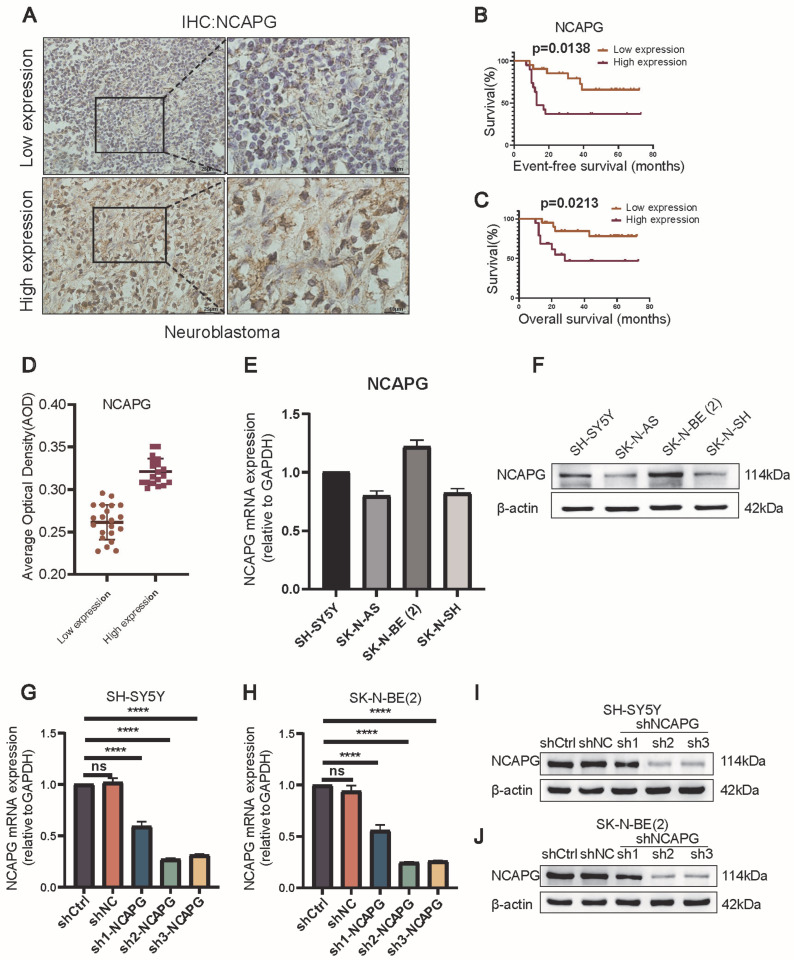
NCAPG expression in neuroblastoma tissues and cell lines. (**A**) Typical photograph of NCAPG protein staining intensity in neuroblastoma tissues. (**B**,**C**) Significant variations in EFS (*p* = 0.0138) and OS (*p* = 0.0213) between neuroblastoma patients with high and low NCAPG expression. (**D**) AOD of neuroblastoma tissues. (**E**,**F**) Levels of *NCAPG* mRNA relative to GAPDH and protein expression in SH-SY5Y, SK-N-AS, SK-N-BE (2), and SK-N-SH cells. (**G**,**H**) qRT-PCR to confirm *NCAPG* mRNA expression relative to GAPDH in lentivirus-transfected neuroblastoma cells. (**I**,**J**) Western blotting to detect the NCAPG protein expression in lentivirus-transfected neuroblastoma cells. Data are expressed as the mean ± SD for three independent experiments. ns: no significance, **** *p* < 0.0001.

**Figure 5 ijms-24-14946-f005:**
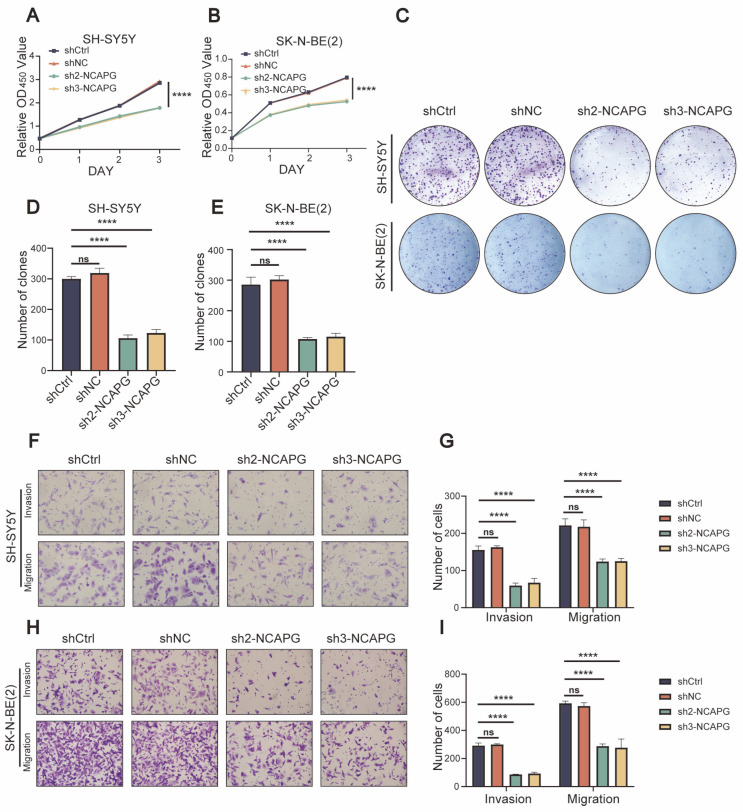
Effect of *NCAPG* depletion on the proliferation, migration, and invasion capabilities of neuroblastoma cells. (**A**,**B**) Proliferation abilities as measured via CCK-8 in neuroblastoma cells following *NCAPG* gene interference. (**C**–**E**) Colony formation capacity as measured via colony formation assay in neuroblastoma cells following *NCAPG* gene interference. Scale bar, 250µm. (**F**–**I**) Migration and invasion abilities as measured via transwell assay in neuroblastoma cells following *NCAPG* gene interference. Scale bar, 10µm. Data are expressed as the mean ± SD for three independent experiments. ns: no significance, **** *p* < 0.0001.

**Figure 6 ijms-24-14946-f006:**
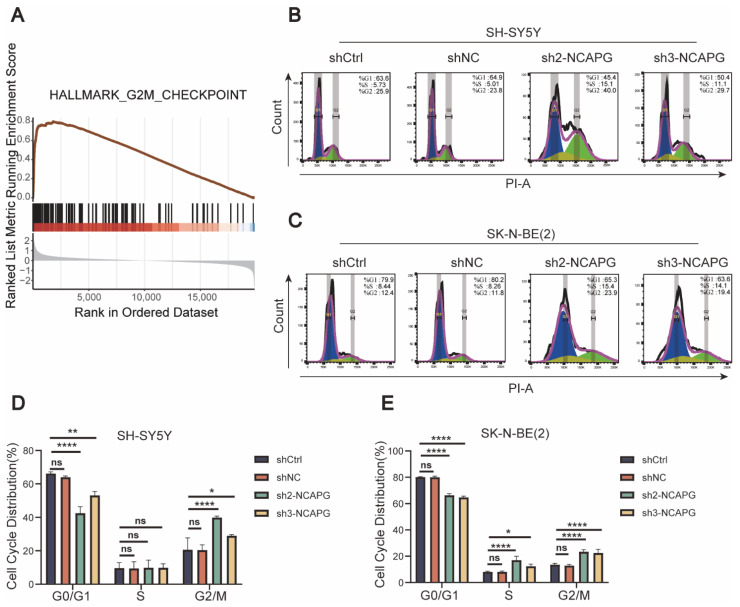
Effect of *NCAPG* depletion on the cell cycle of neuroblastoma cells. (**A**) GSEA analysis indicates a significant correlation between increased *NCAPG* expression and the G2M checkpoint. (**B**,**C**) *NCAPG* knockdown cells’ cell cycle development was recorded using flow cytometric analysis. (**D**,**E**) Cell cycle distribution in *NCAPG* knockdown cells. Data are expressed as the mean ± SD for three independent experiments. ns: no significance, * *p* < 0.05, ** *p* < 0.01, **** *p* < 0.0001.

**Figure 7 ijms-24-14946-f007:**
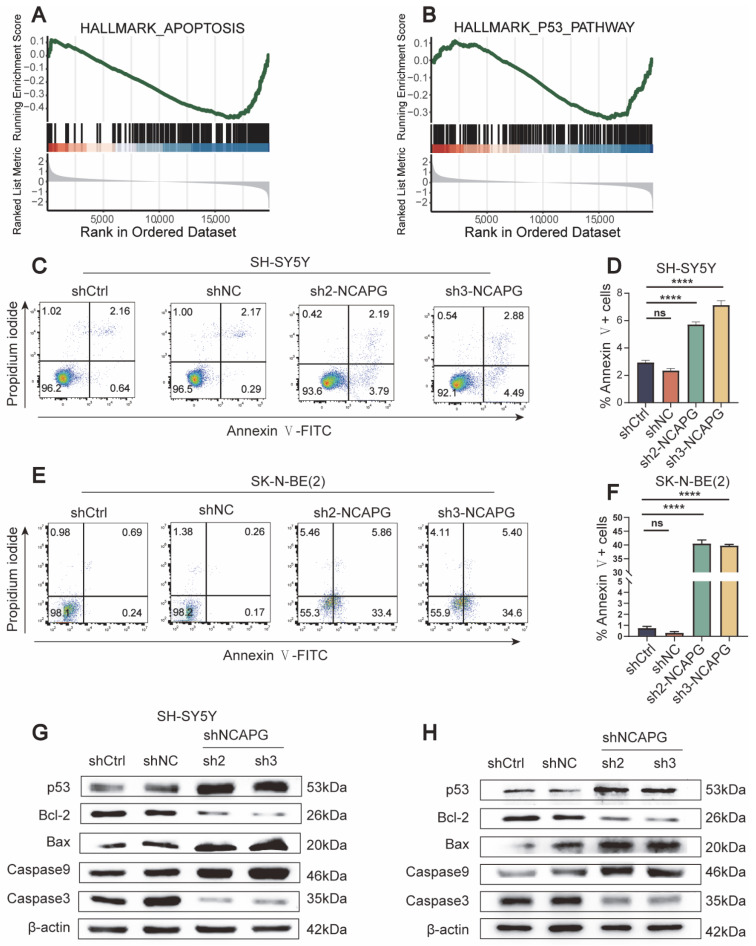
Effect of *NCAPG* depletion on the apoptosis of neuroblastoma cells. (**A**) GSEA analysis demonstrated that lower *NCAPG* expression is linked to apoptosis. (**B**) GSEA analysis demonstrated that decreased *NCAPG* expression is linked to the *P53* pathway. (**C**–**F**) The percentage of *NCAPG* knockdown cells that underwent apoptosis was measured using an Annexin V/PI staining assay. (**G**,**H**) Western blotting was performed to quantify p53, Bcl-2, BAX, Caspase9, and Caspase3 protein levels after *NCAPG* was silenced in neuroblastoma cells. Data are expressed as the mean ± SD for three independent experiments. ns: no significance, **** *p* < 0.0001.

**Table 1 ijms-24-14946-t001:** Patients’ characteristics and their association with NCAPG expression.

Characteristic NCAPG Expression	Low Expression (*N* = 21), *N* (%)	High Expression (*N* = 19), *N* (%)	*p*-Value
Gender			0.796
Male	8 (50)	8 (50)	
Female	13 (54)	11 (46)	
Age			0.115
<18	8 (73)	3 (27)	
≥18	13 (45)	16 (55)	
INSS stage			0.004
1/2/4s	14 (78)	4 (22)	
3/4	7 (32)	15 (68)	
COG risk			0.004
LR/IR	14 (78)	4 (22)	
HR	7 (32)	15 (68)	
Bone marrow metastasis			0.004
No	17 (71)	7 (29)	
Yes	4 (25)	12 (75)	
Distant metastasis			0.027
No	14 (70)	6 (30)	
Yes	7 (35)	13 (65)	
Histological type			0.062
Ganglioneuroblastoma	9 (75)	3 (25)	
Neuroblastoma	12 (43)	16 (57)	
MYCN amplification			0.112
No	17 (61)	11 (39)	
Yes	4 (33)	8 (67)	

**Table 2 ijms-24-14946-t002:** Univariate and multivariate analysis of NCAPG expression and neuroblastoma patient survival.

Subtype	Univariate Analysis		Multivariate Analysis	
	Hazard Ratio (95%CI)	*p*-Value	Hazard Ratio (95%CI)	*p*-Value
Gender				
Male	1			
Female	1.628 (0.500–5.301)	0.419		
Age				
<18	1			
≥18	2.328 (0.514–10.542)	0.273		
INSS stage				
1/2/4s	1		1	
3/4	15.302 (1.974–118.632)	0.009	3.379 (0.354–32.263)	0.290
Distant metastasis				
No	1		1	
Yes	21.972 (2.788–173.127)	0.003	10.492 (1.049–104.894)	0.045
Histological type				
Ganglioneuroblastoma	1			
Neuroblastoma	1.137 (0.349–3.701)	0.832		
MYCN amplification				
No	1			
Yes	2.073 (0.674–6.373)	0.203		
NCAPG expression				
Low	1		1	
High	3.636 (1.110–11.913)	0.033	2.639 (0.741–9.391)	0.134

## Data Availability

The publicly available data are provided in DepMap, GEO (GSE49710) and ArrayExpress (E-MTAB-8248) databases.

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
