# Peer review of "Identification of NCAPG as an Essential Gene for Neuroblastoma Employing CRISPR-Cas9 Screening Database and Experimental Verification"

_ijms, 2023, doi:10.3390/ijms241914946_

Round 1

Reviewer 1 Report

The manuscript presented here contains interesting information obtained from mining existing databases. However, currently, crucial information regarding representativeness is missing, and therefore I cannot judge the merit of this work. Happy to re-review after authors add this information.

Major

-          Figure 4, 5, 6, 7 – how many independently performed experiments with how many independent replicates are included in these figures? This information is currently missing in the figure legends and needs to be added before review can be completed, as representability can now not be checked.

-          Figure 7E looks poorly compensated to me. Unlike the figure above, here, the cell populations are pulling towards the Y axis, hinting at improper compensation. I urge authors to look at this very carefully.

-          General – I am missing a flow cytometry gating strategy – please add this to a (new) supplemental figure.

Minor

Line 96 – please define MCODE

Figure 4E – relative to what? GAPDH – please state so preferably in the axis, but at least in the figure legend.

Figure 4 GH/IJ – mRNA comes before protein – I would suggest to swap these panels around.

Line 218 – please add a distinction regarding the differences between the cell lines – both were not equally susceptible to induction of apoptosis/cell death as indicated by the significant but marginal increase in Annexin-V+ cells in the SH-SY5Y cell line.

No specific comments yet, as I didn't do a full review due to the missing information.

Reviewer 2 Report

The researchers have proposed that they have identified a new candidate gene, NCAGP, which has clinical and biological relevance in neuroblastoma. At first glance, this seems interesting. However, on a thorough read, there are multiple issues with the paper. Citations are either not provided, incorrect, or out of date. Most commonly, they are not provided where they need to be. Many analyses are poorly described, incorrectly cited, or incorrectly described (e.g., multivariate analysis). The sources of tissues and properties of those tissues are poorly described. All these issues lead one to question the validity of the results. After significant editing, publication could be considered, but in its current state the paper lacks appropriate detail and would lead to retraction. 

Some of the syntax and citations are incorrect. For example, in the introduction is says "over 50% of those diagnosed with high risk neuroblastoma encounter development or relapse despite receiving intensified treatment regimens and immunotherapy." That is unclear. In introduction, the citation to 5-year overall survival is linked to a paper on relapsed disease, which is both not the correct citation and also quite dated (12 years old at this point). 

The authors state that TP53 deletion is part of the oncogenesis of neuroblastoma. TP53 is rarely deleted in neuroblastoma; more commonly, MDM2 is activated suppressing TP53 activity. The citation they use states that. 

The introduction references DepMap without explaining it at any depth (tumors vs cell lines, method of evaluation, validation). There is no citation related to DepMap itself. It needs a citation and greater explanation. 

There is a reference to a "neuroblastoma dataset" in the introduction and the methods. While citation of a GSE dataset is OK, it is better to cite the paper in which the dataset was used, so the original authors of that dataset can be credited. The citations used also reference methods papers and not information on the databases or datasets. 

Abbreviations like CERES score and WGCNA need to be explained at first statement, not in the methods. These methods also need to be properly cited; the citation for CERES is not a complete explanation, and the WGCNA is not cited at all. 

It should be made clear that the data from 2.1 are all in silico analyses with extrapolation of function and interactions from prior studies, most of which are not in neuroblastoma, and thus these are all largely theoretical. 

The explanation of the risk score in 2.2 is unclear. First it says 10 genes were used, but then it shows a formula with only 2 genes. LASSO-COX should also be cited in the results, not just in the methods. Otherwise the reader has to hunt for how the research was performed. 

Which cohorts were used for which analyses is very hard to discern and poorly cited. In any case, the analyses that yield the Kaplan Meier curve in figure 2 are effectively no better than current approaches using stage and MYCN amplification. This really dampens the enthusiasm for the work. These analyses should have been done using high-risk tumor datasets only, and/or with subanalyses of high-risk vs all tumors. 

Figure 3 shows really no significance of NCAPG expression vs current biomarkers. Figure 3B is not a multivariate analysis; it is univariable analysis of different markers in the same dataset. Multivariable analysis would control for the independent effects of each variable. That is not what is shown. 

The experimental studies on the neuroblastoma cell lines (which are generally highly sensitive cell lines) are superficial at best. Knockdown by shRNA is unclear, as the blots shown do not really seem to match the percentages shown in the graphs in Figure 4. The follow-up studies are simplistic - proliferation, cell cycle analysis and apoptosis. Nothing is performed to clarify any actual function of the candidate protein. 

With these issues, the results have very low impact. Additional work is needed. 

Adequate, though some syntax could be improved

Round 2

Reviewer 1 Report

I would like to thank the authors for addressing my comments. I have no further comments, except for the minor two below which can be corrected during proofing of the manuscript.

Figure 4D – there is a space between Averg and e on the axis of the figure, please change upon typesetting.

Authors, please note that you refer to figure 7F on line 285, which doesn’t exist, please change to E upon typesetting.

Author Response

Thank you very much for your valuable comments and suggestions on our manuscript entitled “Identification of NCAPG as an essential gene for neuroblastoma employing CRISPR-Cas9 screening database and experimental verification” (ijms-2533003). Following the reviewers’ comments, we have modified and improved our manuscript according to your kind advice and referee’s detailed suggestions. We sincerely hope this manuscript will be acceptable to be published on International Journal of Molecular Sciences.

  1. Figure 4D – there is a space between Averg and e on the axis of the figure, please change upon typesetting.

Response: We thank the reviewer for pointing this out. We have modified the word in Figure 4D (Page 9, Figure 4D).

  1. Authors, please note that you refer to figure 7F on line 285, which doesn’t exist, please change to E upon typesetting.

Response: We highly appreciate your kind reminder. We refer to Figure7C-D, Figure 7E-F again according to your comments. (Page 13, lines 264-265)

Special thanks to you for your kind comments.

We would like to take this opportunity to thank you for all your time involved and this great opportunity for us to improve the manuscript. We hope you will find this revised version satisfactory. 

With best wishes,

Yours sincerely,

Qiang Zhao

Sep 07, 2023

Department of Pediatric Oncology, Tianjin Medical University Cancer Institute and Hospital, Huan-Hu Xi Road, Ti-Yuan-Bei, He xi District, Tianjin 300060, China.

Email: zhaoqiang@tjmuch.com.

Reviewer 2 Report

I appreciate the efforts by the researchers to address the concerns with their original submission regarding their work on NCAPG as a key node in the biology of neuroblastoma. They have taken a focused effort to address the specific concerns where possible, including updates on citations and improved explanations of many of the "dry lab" methods used. 

That said, there remains a few key issues. 

1) The researchers performed biostatistical approaches to evaluate the value of NCAPG expression as it correlates with outcomes as based on the Fischer dataset. However, they used the entire dataset. Within that dataset are patients with both low (including likely low and intermediate-risk tumors as per the INRG stratification) and high-risk tumors. As a result, their initial calculations resulted in not only identifying a gene with differential expression between two groups that can be readily identified with current clinical markers, but that also perhaps identified a cutpoint that has insufficient clinical impact. It would have been far more impactful to narrow the dataset to the high-risk tumors only, then identify key genes and the appropriate cutpoint. 

Analysis after the fact, shown in Supplementary Figure 3, doesn't really fix this problem, as it is using a tool defined by low-and-high-risk tumors on just high-risk tumors. Furthermore, these survival analyses were used simply with median or mean cutpoint, but in a validation set you should use cutpoints defined by your training set (there is no way for the same cutpoint to generate equal patient cohorts in the validation sets). A different cutpoint may establish better utility of the gene expression tool (e.g., 1/3 of patients with high-risk disease may have higher expression of gene X, and all die of disease, but 2/3 have lower expression and almost all survive). 

Incidentally, using the R2 software platform, I was able to identify NCAGP as one of the top five genes correlated with overall survival (poor outcomes) using their online tools in ~4 minutes (8 including the re-analysis of high-risk tumors). In those high-risk tumors, patients did poorly regardless of NCAGP, just at different rates. 

2) The researchers try to establish that NCAGP expression is independent of other markers, but then show in Figure 3 how they actually correlate with other markers. This is somewhat confusing.

3) In figure 4, the knockdown of NCAGP doesn't really match between the western blots and the graphs. Was there a reason CRISPR-Cas9 couldn't have been used to simply eliminate expression, either entirely or conditionally?

4) I appreciate the additional gene expression profiling of the effects of NCAGP modulation. However, none of these effects are necessarily direct. Knockdown of a gene with functions on chromosome stabilization would be expected to trigger DNA damage repair pathways and apoptosis. 

5) The largest remaining issue is that the value of future study of NCAGP is not clear. As a biomarker, it's not likely to change paradigms of clinical risk stratification. Researchers have been trying for 20+ years to define gene expression profiles of ultra-high-risk disease without more success than currently available. While studies into NCAGP in the lab may elucidate some aspects of NBL biology, it is not a likely tractable target therapeutically unless the researchers demonstrate that the expression of the protein from the gene is particularly upregulated in NBL as opposed to other healthy tissues, allowing a therapeutic window in which attack of that protein would spare effects on healthy cells. Given its inherent function, that seems unlikely but it could be shown and added to the paper (without much effort). 

Seems overall ok, just needs a good copyedit. 

Author Response

Thank you very much for your comments and professional advice on our manuscript entitled “Identification of NCAPG as an essential gene for neuroblastoma employing CRISPR-Cas9 screening database and experimental verification” (ijms-2533003). These opinions help to improve academic rigor of our article. Based on your suggestion and request, we have made corrected modifications on the revised manuscript. We hope that our work can be improved again. Furthermore, we would like to show the details as follows:

  1. The researchers performed biostatistical approaches to evaluate the value of NCAPG expression as it correlates with outcomes as based on the Fischer dataset. However, they used the entire dataset. Within that dataset are patients with both low (including likely low and intermediate-risk tumors as per the INRG stratification) and high-risk tumors. As a result, their initial calculations resulted in not only identifying a gene with differential expression between two groups that can be readily identified with current clinical markers, but that also perhaps identified a cutpoint that has insufficient clinical impact. It would have been far more impactful to narrow the dataset to the high-risk tumors only, then identify key genes and the appropriate cutpoint. Analysis after the fact, shown in Supplementary Figure 3, doesn't really fix this problem, as it is using a tool defined by low-and-high-risk tumors on just high-risk tumors. Furthermore, these survival analyses were used simply with median or mean cutpoint, but in a validation set you should use cutpoints defined by your training set (there is no way for the same cutpoint to generate equal patient cohorts in the validation sets). A different cutpoint may establish better utility of the gene expression tool (e.g., 1/3 of patients with high-risk disease may have higher expression of gene X, and all die of disease, but 2/3 have lower expression and almost all survive). Incidentally, using the R2 software platform, I was able to identify NCAGP as one of the top five genes correlated with overall survival (poor outcomes) using their online tools in ~4 minutes (8 including the re-analysis of high-risk tumors). In those high-risk tumors, patients did poorly regardless of NCAGP, just at different rates. 

Response: Thank you for your kind comments!

We used the entire GSE49710 dataset to screen essential genes for the  prognosis of neuroblastoma patients and identified NCAPG as the critical gene. We fully appreciate your comment that the prognostic value of NCAPG may derive from its differential expression between high-risk and low-risk groups, as the established risk classification system has been widely recognized by the Children’s Oncology Group (COG) and the International Society of Paediatric Oncology (SIOP), two major guidelines used for the current management of pediatric cancer worldwide. In other words, genes that are differentially expressed in the high-risk and low-risk groups could show strong potential to demonstrate significant prognostic value. As you mentioned, it would have been far more impactful to narrow the dataset to the high-risk tumors only and identify essential genes and the appropriate cut-point. Regrettably, limited by the low sample size and poor prognosis of the high-risk group, identification of key prognosis-related genes using single transcriptome sequencing data in the high-risk group maybe challenging. Multi-omics data integration analysis is a promising direction, and we will focus on high-risk groups in future studies.

Furthermore, thanks for your interest in the cut-point! A consistent cut-point does contribute to expanding the applicability and accuracy of the risk signature. According to your comment, we used the median of the training group risk score as the cut-point for the validation groups. More than half of the patients in the internal and external validation datasets were reclassified into the low-risk group. As expected, the cut-point derived from the training group showed comparable prognostic value in the validation groups (Revised Figure 1 A-C). We also tested the prognostic value of NCAPG in the high-risk group in the R2 platform. Although the high-risk group presented a poor prognosis, NCAPG could still distinguish two entities with distinct prognoses in this group (Revised Figure 1 D-E). These results suggest the potential prognostic value of NCAPG in high-risk neuroblastoma, and further research is still necessary.

Revised Figure 1. The prognostic value of the risk signature and NCAPG expression. (A-C) The Kaplan-Meier analysis of the risk signature in the train dataset (A), internal validation dataset (B), and external validation dataset (C). (D-E) The Kaplan-Meier analysis of NCAPG expression in the GSE62564 dataset (D) and TARGET dataset (E) in the R2 platform.

  1. The researchers try to establish that NCAGP expression is independent of other markers, but then show in Figure 3 how they actually correlate with other markers. This is somewhat confusing.

Response: Thank you very much for your interesting comments!

NCAPG expression was indeed related to multiple clinical characteristics, such as MYCN status, COG risk, INSS stages, and Tumor progression. However, this correlation did not contradict its role as an independent prognostic factor. The remarkable correlation of NCAPG expression with clinical characteristics suggest that it may have a similar prognostic value to these established prognostic factors.  Based on the univariate analysis, we found that NCAPG exhibited significant prognostic value. In subsequent multivariate analysis incorporating multiple prognostic factors, NCAPG retained significant prognostic value. This result suggested that the prognostic value of NCAPG did not derive from other prognostic factors, and it could be identified as an independent prognostic factor. The conclusion was not only obtained from the the test dataset but also validated in the tumor samples in our hospital.

  1. In figure 4, the knockdown of NCAGP doesn't really match between the western blots and the graphs. Was there a reason CRISPR-Cas9 couldn't have been used to simply eliminate expression, either entirely or conditionally?

Response: Thanks for your kind suggestions, which are valuable for improving the manuscript's accuracy.

We performed qRT-PCR and WB experiments to verify the synchronous decrease of NCAPG mRNA and protein expression levels in both neuroblastoma cell lines after knockdown by shRNA (Page 9, Figure 4G-J).

Actually, we tried to knock out NCAPG using CRISPR-Cas9 technology, but all neuroblastoma cells gradually died after lentivirus infection, due to the lethality of NCAPG, it was impossible to construct a stable NCAPG knockout cell lines. Besides,we downloaded the CERES scores of NCAPG in neuroblastoma cells from the DepMap database. We found that they were all less than -1, indicating that neuroblastoma cells highly depended on this gene (Revised Figure 2). Meanwhile, we have yet to find a successful NCAPG stable knockout cell line construction case. So, in our research, we can only use NCAPG shRNAs to knock down the expression of NCAPG.

Revised Figure 2. The CERES scores of NCAPG in neuroblastoma cells.

  1. I appreciate the additional gene expression profiling of the effects of NCAGP modulation. However, none of these effects are necessarily direct. Knockdown of a gene with functions on chromosome stabilization would be expected to trigger DNA damage repair pathways and apoptosis. 

Response: Thank you for the constructive comments and suggestions.

We couldn't agree more with you. In this study, we can observe that neuroblastoma cells undergo significant apoptosis after NCAPG is knocked down. Therefore, flow cytometry demonstrated that the proportion of apoptotic cells increased significantly after NCAPG knockdown. Meanwhile, we detected the changes in apoptosis-related proteins such as Bcl-2, Bax, and Caspase family. We used GSEA analysis to show a significant correlation between NCAPG and the P53 signaling pathway, and then we used Western blotting to prove the changes in P53 protein after NCAPG knockdown. We are very sorry that we did not further investigate the direct relationship between NCAPG and the P53 pathway. An in-depth study of the potential mechanisms by which NCAPG promotes apoptosis will be more helpful to our understanding of this gene, and we will perform this in our future study. Thank’s again for pointing out the shortcomings and limitations of our research, and your suggestions provide essential guidance for our subsequent further research.

  1. The largest remaining issue is that the value of future study of NCAGP is not clear. As a biomarker, it's not likely to change paradigms of clinical risk stratification. Researchers have been trying for 20+ years to define gene expression profiles of ultra-high-risk disease without more success than currently available. While studies into NCAGP in the lab may elucidate some aspects of NBL biology, it is not a likely tractable target therapeutically unless the researchers demonstrate that the expression of the protein from the gene is particularly upregulated in NBL as opposed to other healthy tissues, allowing a therapeutic window in which attack of that protein would spare effects on healthy cells. Given its inherent function, that seems unlikely but it could be shown and added to the paper (without much effort). 

Response: Thanks for your comments, which are highly appreciated.

We quite agree with your points and suggestions. Neuroblastoma mainly arises from the adrenal gland, in order to preliminary explore the expression of NCAPG between normal and tumor tissues, we obtained normal adrenal tissue sequencing data from the GTEx database and neuroblastoma tissue(adrenal origin) sequencing data from the TARGET database, and found that the expression of NCAPG was significantly up-regulated in neuroblastoma tissues (Revised Figure 3). However, further validation is needed in the cell level, normal adrenal, and tumor tissue. In the future study, we will collect healthy adrenal glands and paired neuroblastoma tumor tissue originating from the adrenal gland to validate the expression of NCAPG in different groups, and aslo explore the difference between healthy cell and neuroblastoma cells.

Besides,we also revised the discussion section in the manuscript. The end of the discussion has been revised as “Although NCAPG showed significant prognostic value in our study, its role in improving the established risk stratification system still needs to be verified in large cohort clinical research. In addition, normal cells also require NCAPG to maintain their normal biological functions, which makes targeted therapy against NCAPG challenging, and it is crucial to find an appropriate therapeutic window. Our subsequent work would continue to concentrate on the possibility of NCAPG in neuroblastoma clinical management and further mechanistic investigations to explore potential therapeutic targets.” (Page 16, line 369-376).

Revised Figure 3. The expression of NCAPG in normal adrenal and neuroblastoma tissue.

We tried our best to improve the manuscript and made some changes marked in yellow in revised paper which will not influence the content and framework of the paper. We appreciate for your warm work earnestly, and hope the correction will meet with approval. Once again, thank you very much for your comments and suggestions.

We would like to thank the referee again for taking the time to review our manuscript.

With best wishes,

Yours sincerely,

Qiang Zhao

Sept 07, 2023

Department of Pediatric Oncology, Tianjin Medical University Cancer Institute and Hospital, Huan-Hu Xi Road, Ti-Yuan-Bei, He xi District, Tianjin 300060, China.

Email: zhaoqiang@tjmuch.com.
